# AbundanceR: A Novel Method for Estimating Wildlife Abundance Based on Distance Sampling and Species Distribution Models

**Xinhai Li** [1,2,*,†] **, Ning Li** [3,†] **, Baidu Li** [4]**, Yuehua Sun** [1] **and Erhu Gao** [5]

1   Key Laboratory of Animal Ecology and Conservation Biology, Institute of Zoology, Chinese Academy of Sciences, Beijing 100101, China; sunyh@ioz.ac.cn
2   College of Life Sciences, University of Chinese Academy of Sciences, Beijing 100049, China
3   Institute of Applied Ecology, Nanjing Xiaozhuang University, Nanjing 211171, China; lining@njxzc.edu.cn
4   Lassonde School of Engineering, York University, 4700 Keele Street, Toronto, ON M3J 1P3, Canada; baidu@my.yorku.ca
5   Academy of Inventory and Planning, National Forestry and Grassland Administration, Beijing 100714, China; gaoerhu2022@126.com
*   Correspondence: lixh@ioz.ac.cn; Tel.: +86-10-64807898
†   These authors contributed equally to this work.

**Abstract:** Appropriate field survey methods and robust modeling approaches play an important role in wildlife protection and habitat management because reliable information on wildlife distribution and abundance is important for conservation planning and actions. However, accurately estimating animal abundance is challenging in most species, as usually only a small proportion of the population can be detected during surveys. Species distribution models can predict the habitat suitability index, which differs from species abundance. We designed a method to adjust the results from species distribution models to achieve better accuracy for abundance estimation. This method comprises four steps: (1) conducting distance sampling, recording species occurrences, and surveying routes; (2) performing species distribution modeling using occurrence records and predicting animal abundance in each quadrat in the study area; (3) comparing the difference between field survey results and predicted abundance in quadrats along survey routes, adjusting model prediction, and summing up to obtain total abundance in the study area; (4) calculating uncertainty from three sources, i.e., distance sampling (using detection rate), species distribution models (using R squared), and differences between the field survey and model prediction [using the standard deviation of the ratio (observation/prediction) at different zones]. We developed an R package called abundanceR to estimate wildlife abundance and provided data for the Tibetan wild ass (*Equus kiang*) based on field surveys at the Three-River-Source National Park, as well as 29 layers of environmental variables covering the terrestrial areas of the planet. Our method can provide accurate estimation of abundance for animals inhabiting open areas that can be easily observed during distance sampling, and whose spatial heterogeneity of animal density within the study area can be accurately predicted using species distribution models.

**Keywords:** abundance; distance sampling; population density; R package; species distribution models; wildlife survey; uncertainty

## 1. Introduction

Species abundance is fundamental information for ecological research, and various methods are used for its estimation [1–3]. Direct counting, including spotlight, track, and roadkill counts, can provide initial information on the abundance of several species, such as water birds and deer [4,5]. Catch-per-unit effort is often used for regularly captured species, such as commercial fish [6,7]. The mark-recapture technique has been widely used and it is suitable for populations in a small region when target species can be easily captured [8,9].

Camera-based models are playing an important role in abundance estimation in recent years [10,11]. The spatially explicit capture-recapture technique uses the camera trap data of species with identical characteristics and can provide good abundance estimation [12,13]. If individuals of the target species are not distinguishable, the random encounter model [14], more advanced time-to-event estimates and space-to-event estimates [10], and the movement-based method [15] are appropriate for abundance estimation. Mark-recapture and camera traps are appropriate for small regions.

At a large spatial scale, distance sampling [16] or regular line transects are suitable for species that can be directly observed, yet the results cannot be expanded to unsurveyed areas [16,17] because species are not evenly distributed [18]. The advantage of distance sampling is that it estimates the detection function, quantifying the relationship between the probability of detection and the animal-to-observer distance so that it provides a measure of survey uncertainty. The N-mixture model can quantify both the detection rate (using repeated surveys) and the contribution of environmental variables so that it can be extended to unsurveyed areas [19], yet it is not capable of handling a large number of environmental variables, which limits its ability for abundance estimation.

Species distribution models (SDMs) have been widely used to estimate species distribution in unsurveyed areas [20], as some SDMs can quantify the association of species occurrences and a large number of environmental variables [21,22] so as to predict animal density at unsurveyed areas. Boyce and McDonald [23] suggested that animal abundance could be estimated by summing the probabilities of presence calculated by SDMs, but such estimations are inaccurate unless the population is at carrying capacity or in an ideal free distribution [24]. In most cases, species distribution is constrained by environmental variables, which can be estimated by SDMs, yet animal populations are seldom at equilibrium nor have an ideal-free distribution [24]. Currently, there is no reliable method for estimating the abundance of animals occurring in a large region where only a small part is surveyed.

To estimate animal abundance in large areas, researchers usually applied specific methods that were only suitable for their target species. For example, Stauffer et al. used home range size and mean group size for the abundance estimation of wolves (*Canis lupus*), which was a scale-up process from occupancy to abundance [25]. Santos et al. used nesting beach monitoring data such as nest counts and clutch frequency to estimate the abundance of marine turtles [26]. Teton et al. used natural markings (no ear-tags or neck-bands) to identify individuals for mark-resight population estimation of invasive wild pigs (*Sus scrofa*) [27]. Shertzer et al. estimated abundance of an open population, gray triggerfish (*Balistes capriscus*), by pairing two models, a Markovian mode using data from telemetry tags for movement rate, and a Lincoln-Petersen abundance estimator model modified to account for mortality and movement [28]. Howard et al. used age-at-harvest data and auxiliary information such as estimated survival rates, harvest probabilities, and hunter effort to estimate mountain lion (*Puma concolor*) abundance in Arizona, USA [29].

In this study, we developed a more general method to estimate species abundance in a large region based on distance sampling and SDMs. We used SDMs to quantify the heterogeneity of spatial distribution to estimate animal abundance in unsurveyed areas and compare the SDM-predicted abundance on survey routes with field survey results with the aim of adjusting model prediction. The uncertainty of distance sampling, species distribution modeling, and spatial heterogeneity of the observation-prediction ratio was considered for the abundance estimation. All algorithms were coded in R [30] and can be installed from the GitHub repository.

## 2. Methods

The data used to estimate animal abundance were obtained from distance sampling. Both species occurrence data and survey routes were required. We provided survey data of the Tibetan wild ass (*Equus kiang*) in the Three-River-Source National Park as an example. Random forest was selected as the SDM for predicting the number of individuals in each quadrat in the entire study area. We compared the SDM prediction with field survey results.

By adjusting the prediction, we obtained a better estimation of animal abundance. To install the package and load it, the following code is used:

```
library(devtools) # or using library(remotes)
install_github("Xinhai-Li/abundanceR")
library(abundanceR)
```

### 2.1. Data Requirements

We used an example dataset to show the data needed to estimate animal abundance using the package abundanceR.

The example data used were from our surveys. We applied a distance sampling protocol [16] to conduct field surveys in the Three-River-Source National Park, which is the central part of the Qinghai-Tibet Plateau, comprising the headwaters of the Yangtze, Yellow, and Lancang rivers, with an area of $960 \times 560$ km$^2$ (92–102° E, 32–37° N) (Figure 1) and an average elevation of approximately 4500 m. We recorded 159 Tibetan wild ass occurrences on the survey routes. The survey results were recorded as GPX or KLM files, including information on species names, counts (number of individuals at one site), vertical distance to observers, latitude and longitude of observers, and date and time (Table 1). Most of the survey routes were on unpaved roads, vehicle speed was 15–30 km/h, and we kept recording while driving.

```
data(kiang) # load the data of distance sampling for the Tibetan wild ass (kiang).
head(kiang) # show the first six rows (Table 1).
```

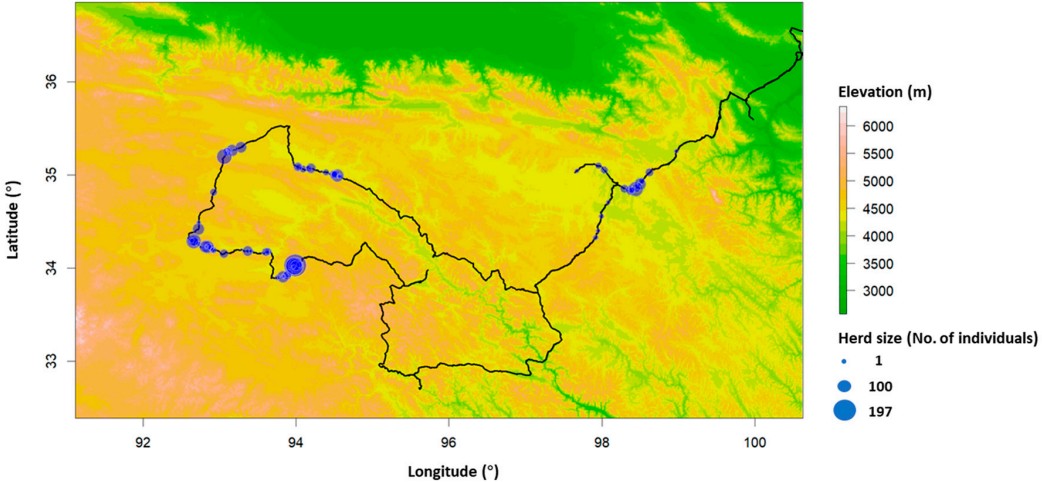

**Figure 1.** The study area in the Three-River-Source National Park. The 159 blue points indicate Tibetan wild ass occurrences of 1039 individuals, and point size indicates species group size ranging from 1 to 197. The black line is the survey route. The background is elevation.

**Table 1.** The survey data (the first six rows) for the Tibetan wild ass (*Equus kiang*) following the distance sampling protocol *.

| Species | Size | Distance | Side | Lat | Lon | Elev | Date | Time |
|---------|------|----------|------|-----|-----|------|------|------|
| kiang | 9 | 130 | e | 34.83078 | 98.37612 | 4217 | 17 July 2017 | 13:28:43 |
| kiang | 32 | 150 | e | 34.84620 | 98.44160 | 4223 | 17 July 2017 | 13:22:53 |
| kiang | 7 | 600 | e | 34.85080 | 98.29750 | 4225 | 17 July 2017 | 13:37:12 |
| kiang | 8 | 350 | e | 34.85908 | 98.45139 | 4232 | 17 July 2017 | 13:18:01 |
| kiang | 1 | 210 | e | 34.87584 | 98.47288 | 4236 | 17 July 2017 | 13:15:24 |
| kiang | 3 | 200 | e | 34.89577 | 98.49666 | 4244 | 17 July 2017 | 13:12:01 |

* In the table, "size" is the group size, i.e., number of individuals at the site (occurrence); "distance" is the distance between the observer and animals; "Side" indicates the direction of the animals from the road; "Elev" is the elevation of the observer (we assumed the animals stay at the same elevation). The names "species", "size", "distance", "Lat" and "Lon" cannot be changed due to case sensitivity. The variables "Side", "Elev", "Date" and "Time" are not needed in the abundance estimation.

## 2.2. Detection Functions in Distance Sampling

We used the R package Distance [31] to estimate the detection function, which quantifies the relationship between the probability of detection and the distance between animals and observers. In the package Distance, three key functions are provided: "hn" is a half-normal function (default), "hr" gives a hazard-rate function, and "unif" provides a uniform function. The package also provides three adjustment terms to tune the detection functions: "cos" gives a cosine term (default), "herm" gives a Hermite polynomial term, and "poly" provides a simple polynomial term. The combination of the three key functions and three adjustment terms is nine, and in abundanceR, we provided a function for all nine combinations and selected the best detection function with the lowest AIC value (Table S1). Based on the selected detection function, we designed to output the average detection rate across the entire range of distances between observers and animals. The average detection rate is an index of survey uncertainty, as the uncertainty decreases with an increase in the detection rate.

The maximum distance that the Tibetan wild ass can be detected is 1500 m, yet we set the truncation distance to 500 m, i.e., the animals outside of that range were ignored. As such, the detection range on each side was 500 m, and the width of the line transect was 1 km; thus, the field observations were comparable with the model prediction at the same spatial scale, i.e., 1-km$^2$ quadrats. The R code for selecting the best detection functions and adjustment terms and estimating the detection probability (Figure 2) is as follows:

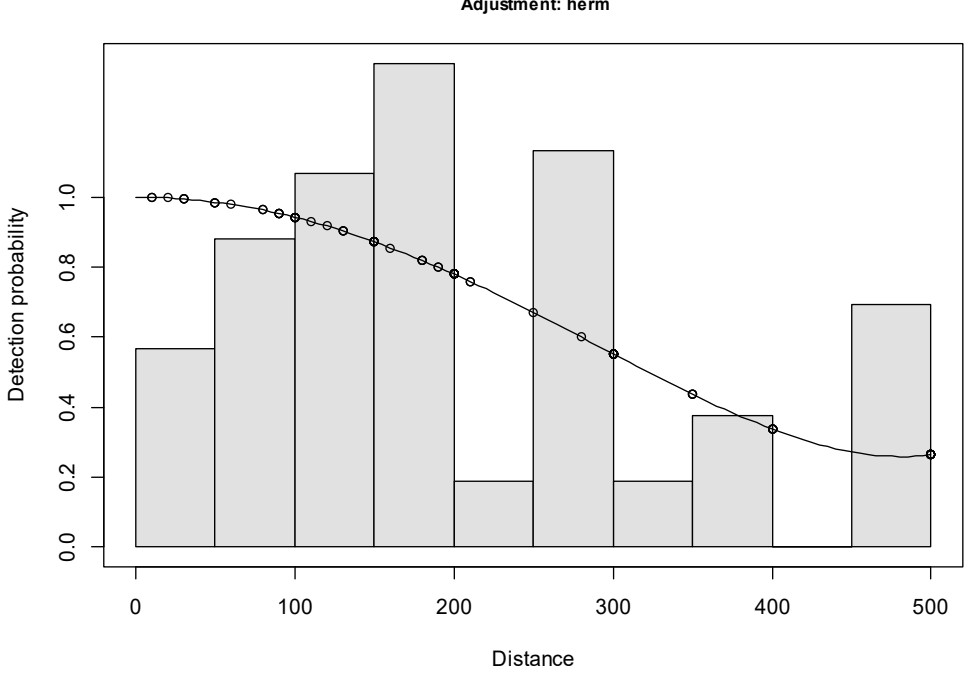

**Figure 2.** The detection function of distance sampling for the Tibetan wild ass in the Three-River Source National Park. The unit of distance is the meter.

```
library(abundanceR) # load the package
# calculate the AIC values for the 9 combinations of detection functions and adjustment terms
AICs = distanceSampling(kiang[kiang$distance <= 500,]) # truncation range is 500 m
AICs = AICs[!is.na(AICs$AIC),] # remove null values in case no AIC value is calculated
# The selected detection function
ds.kiang <- ds(kiang, key = AICs$Key [1], adjustment = AICs$Adjustment [1],
convert.units = 0.001, truncation = 500)
SM = summary(ds.kiang) # the results of distance sampling
Average.p = SM$ds$average.p # average detection rate across the distance range, 0.647
```

survey.uncertainty = 1 − Average.p # survey uncertainty
# Show the detection function (Figure 2)
plot(ds.kiang, main = paste("Key:", AICs$Key [1], "\n", "Adjustment:", AICs$Adjustment [1],
sep = " "))

*2.3. Environmental Variables for Species Distribution Models*

We used 29 environmental variables and animal count data to develop species distribution models. The environmental variables included 19 climate variables [32], elevation [33], human footprint index [34], land cover [35], wetlands [36], as well as solar radiation, wind speed, and water vapor pressure for January and July [32] (Table S2).

All 29 variables were raster layers covering the study area (Figure 1) with a 1-km$^2$ resolution. We compiled data in R. grd format, which is a stack of 29 raster layers. The file size was 69 MB, covering an area of 537,600 km$^2$. The compressed file (BioClim.zip, 23 MB) is too large for a package, so we uploaded it to the author's GitHub repository: https://github.com/Xinhai-Li/abundanceR. To help users access environmental variables for their own study areas, we provided 29 raster layers for all terrestrial regions in the world (Figure S1). The file size was 105 GB, and the compressed size was 8 GB. The file (var29.zip) can be downloaded from the Baidu Cloud Disk$^{TM}$ for users in Mainland China at https://pan.baidu.com/s/1noU8A7WcsuYx0MSiQq6CeQ (access code: 1234); and can also be downloaded from the Google Drive at (https://drive.google.com/drive/folders/1bNh4SdikmjrOkgqE5VOVo86SozD2hvmT?usp=sharing). Users can use the cropLayers function to obtain environmental variables for any terrestrial area on Earth as follows:

library(raster)
BioClim <- brick('var29.grd') # load the 29 environmental variables
data(kiang); head(kiang) # load the species occurrences
# crop the global data to fit users' study area
BioClim = cropLayers(kiang, buffer = 0.2, Envlayers = BioClim)
# the argument, buffer = 0.2, defines the extents of environmental variables are larger than
# that of species occurrences by 0.2 degree at each side (north, south, east, and west) (Figure 3).

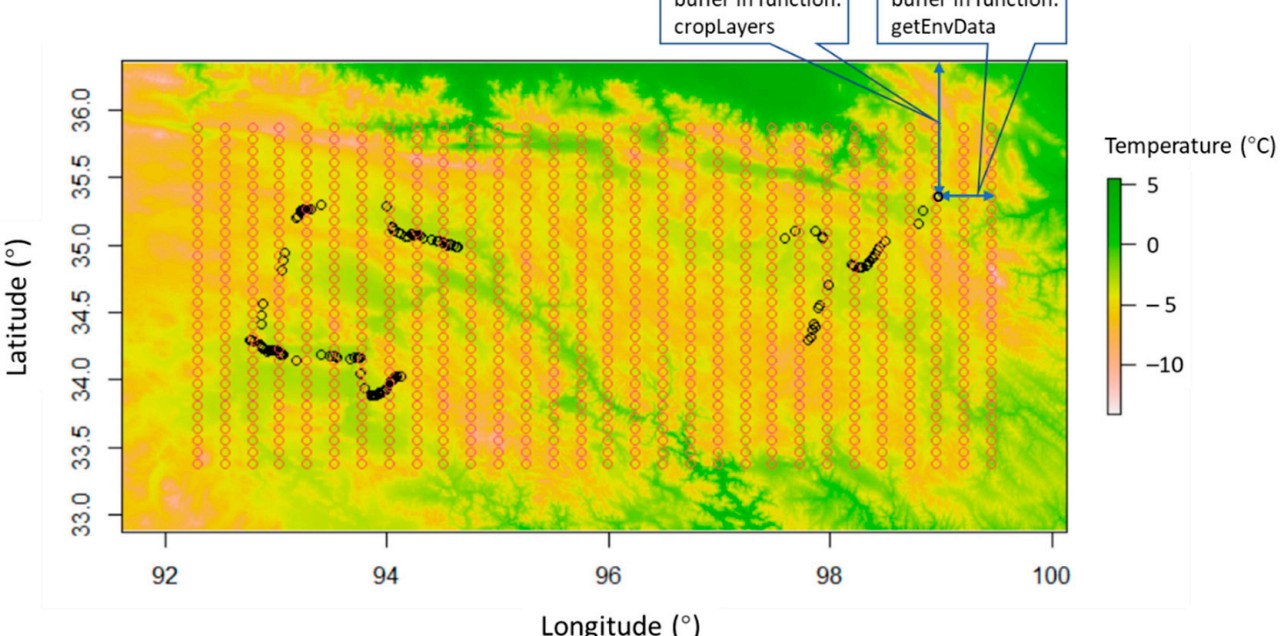

**Figure 3.** The study area (using annual mean temperature as the background), Tibetan wild ass presence data (black circles), and the pseudo-absence points (red circles). The argument buffer is demonstrated for the cropLayers and getEnvData functions.

*2.4. Predicting Species Abundance Using Species Distribution Models*

Among various algorithms, such as the generalized linear model, generalized additive model, support vector machines, random forest, Maxent, multiple adaptive regression spline, and artificial neural networks, we selected random forests because of their high performance [21,37–39].

Random forests require absence data for species distribution modeling. Therefore, we used the getEnvData function to generate pseudo-absence data in the range of occurrences (Figure 3). The dependence variable is the count of the Tibetan wild ass along the survey routes and evenly distributed pseudo-absence data (count 0) in the study area (Figure 3).

```
Data = getEnvData(kiang, buffer = 0.5, absence = 30, Envlayers = BioClim)
# the argument, buffer = 0.5, defines the extents of pseudo-absence points are larger than
# that of species occurrences by 0.5 degree at each side (Figure 3).
# the argument, absence = 30, defines the number of pseudo-absence points is 30*30 = 900
plot(BioClim[[1]], xlab = "Longitude", ylab = "Latitude") # annual mean temperature
points(Data$Lon[Data$Name == "absent"], Data$Lat[Data$Name == "absent"], col = 'red')
points(Data$Lon[Data$Name == "kiang"], Data$Lat[Data$Name == "kiang"])
```

We used the na.roughfix function provided by the randomForest package [40] to replace null values with mean values, because in certain places some variables such as human footprint index has null values while other variables have valid values. The code is: library(randomForest)

```
no.col = ncol(Data) # 33
Dat.fill <- na.roughfix(Data[,2:(no.col-4)]) # no.col-4: use 27 variables. no.col-2: using 29 variables #
    including landcover and wetland
#
# Build the species distribution model
RF <- randomForest(Dat.fill[, 2: ncol(Dat.fill)], Dat.fill[, 1], ntree = 1000,
importance = TRUE, na.action = na.roughfix)
RF # shows the proportion of variance of species count explained by environmental variables.
model.uncertainty = 1-max(RF$rsq) # model uncertainty #0.254
```

The species count can be predicted for every quadrat of the environmental variables, based on the association between species occurrence and environmental variables. Among the 29 environmental variables, land cover and wetland caused unexpected results; therefore, we removed the two variables from the model.

```
# using the model (RF) and environmental data (BioClim) for prediction
pred = popSize(BioClim[[1:27]], RF) # use the first 27 layers
# Show the predicted animal density
plot(pred)
# Change the color
plot(log(1 + log(1 + pred)), xlab = "Longitude", ylab = "Latitude", main = ",
col = colorRampPalette(c("grey90", "green", "yellow", "red"))(12))
# Add species occurrences
points(kiang$Lon, kiang$Lat, pch = 16, cex = log(kiang$size)/2, col = adjustcolor("red", 0.5))
```

The predicted abundance (Figure 4) was log-log transformed to obtain a better color effect. The number of individuals (kiang$Count) varied from 1 to 197; therefore, we used log transformation to compress the difference.

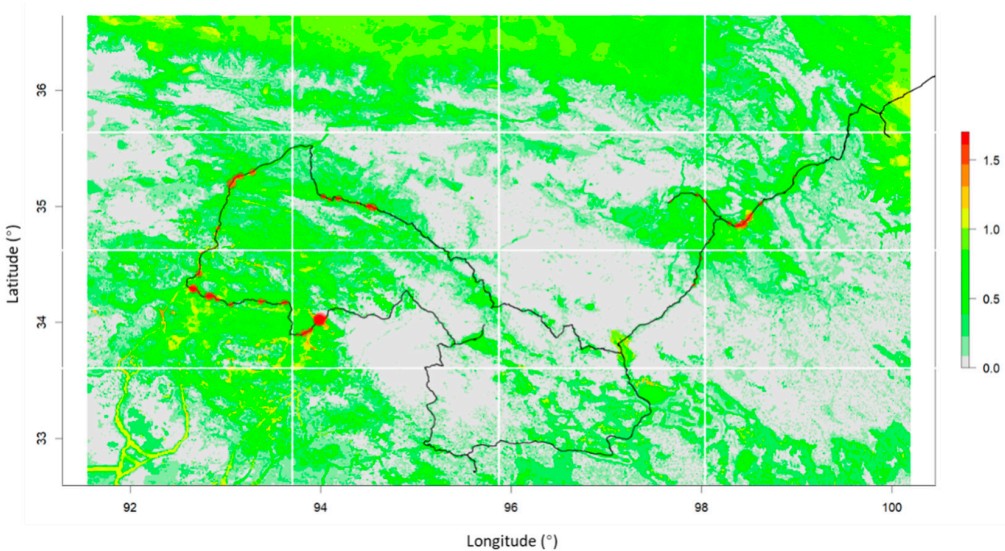

**Figure 4.** The predicted abundance (log-log transformed) of the Tibetan wild ass in the Three-River-Source National Park. The red circles indicate 159 Tibetan wild ass occurrences of 1039 individuals, and circle size indicates species group size ranging from 1 to 197. The black line is the survey route. The study area was divided into 16 zones (separated by white lines), and the observation-prediction ratios were compared among the 16 zones to estimate adjustment uncertainty.

## 2.5. Adjusting Model Prediction

The animal abundance predicted by species distribution models is usually biased [24]. Therefore, we adjusted for model bias using survey results. We loaded the survey route from a GIS shapefile and selected the track points from the route. The track points were 1 km apart (Figure 5). The predicted animal abundance in each quadrat was extracted using the trackPoints function. By comparing the predicted animal abundance along the survey route and the observed number of individuals during the survey, we obtained a ratio of prediction bias, which can adjust the model prediction.

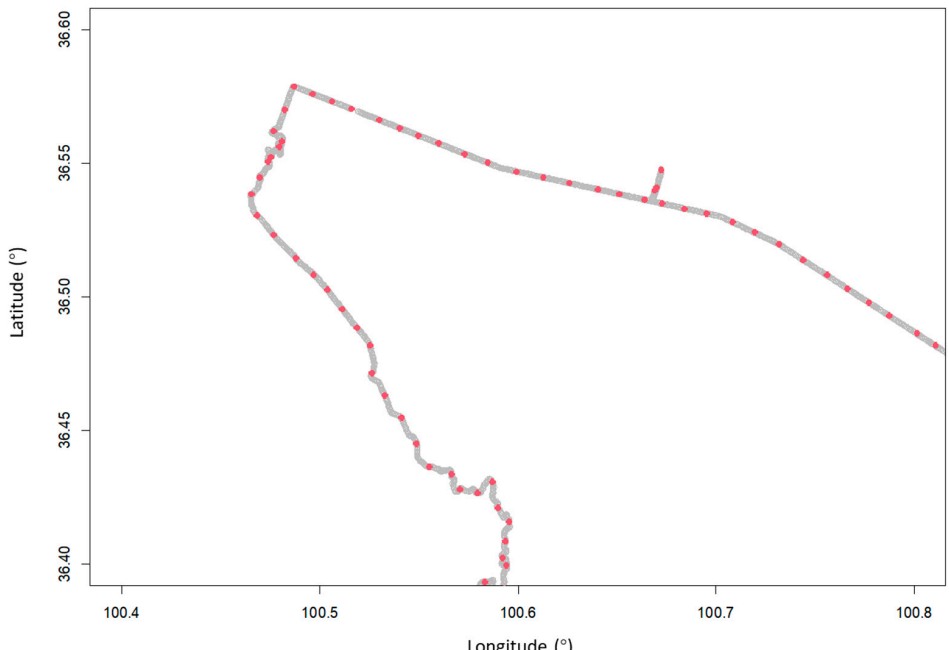

**Figure 5.** One segment of the survey route for the Tibetan wild ass (grey lines) and the selected track points (red points) for extracting predicted animal abundance using the trackPoints function.

The observed animals were also biased because we missed some individuals during the survey. Therefore, we used the average detection rate to adjust for this bias.

```
library(sp)
data(shape); plot(shape) # survey route
tracks = trackPoints(shape) # derive track points with 1 km interval from the survey route
```

The processes of obtaining the original animal abundance estimation, comparing the prediction and observation and making adjustments, are coded below:

```
# Total number of individuals for the raster
pop_ori = cellStats(pred, stat = 'sum', na.rm = TRUE) # 499,705.8
pre <- extract(pred, tracks) # predicted animal abundance on survey routes
pop_pre = sum(pre, na.rm = T) # The predicted number of individuals on the route, 4585
kiang = kiang[kiang$distance <= 500, ] # keep occ within 500 m to match quadrat of 1 km
pop_obs = sum(kiang$size) # observation, 449
pop_obs_adj = pop_obs/detection # distance sampling adjustment, 694
adjust = pop_pre/pop_obs_adj # SDM adjustment, 6.6
# Adjusted animal abundance
pop_est = cellStats(pred, stat = 'sum', na.rm = TRUE)/adjust # 75598
```

We designed a function estPopSize to perform all the above calculations as follows:

```
EST = estPopSize(pred, tracks, kiang, Average.p) # 75598
```

The argument 'pred' is a raster layer of the predicted number of individuals in each quadrat. The argument 'tracks' are the track points on the survey route at 1-km intervals. The argument 'kiang' is a data frame with occurrences of the species recorded during distance sampling. The argument 'Average.p' is the mean detection rate of the distance sampling. This function provides four values: the original model prediction for animal abundance, the predicted animal abundance on the survey route, the animal abundance along the survey route based on field surveys, and the adjusted animal abundance in the study area.

### 2.6. Estimation Uncertainty

To evaluate the uncertainty of the predicted animal abundance, we considered three sources: distance sampling, species distribution modeling, and adjustment based on the ratio of observation and prediction.

Some individuals were missed during distance sampling. Such a situation always occurs, which results in survey uncertainty. Therefore, we used the proportion of missed individuals, which is 1-average-detection-rate, as the index of survey uncertainty. The R code is:

```
survey.uncertainty = 1 − Average.p # survey uncertainty, based on distance sampling
```

Species distribution models depend on the relationship between animal occurrence and environmental variables, i.e., the extent to which environmental variables constrain animal distribution. Accordingly, environmental variables strongly influence habitat specialists and weakly influence habitat generalists. We used the unexplained variance of the dependent variable, the animal counts at their occurrences, as the model uncertainty, which is $1 − R^2$. The ratio of predictions to observations varied in different regions. Therefore, we used the standard deviation of the ratio among regions as the adjustment uncertainty. The R code is:

```
model.uncertainty = 1 − max(RF$rsq) # model uncertainty, based on random forest
```

To calculate the adjustment uncertainty, we divided the survey routes into a number of segments. In the following code, we used argument grid = 4, where the study area was divided into 4 × 4 grids (Figure 4). We calculated the observation-prediction ratio within each grid and obtained the standard deviation of the ratio. The R code is:

adjust.uncertainty = spatialMatch(kiang, tracks, pred, grid = 4)

We assumed that the three indices of uncertainty are independent. Therefore, we multiplied them together as the overall uncertainty:

CI = estimated_abundance * rnorm(1, 1, survey.uncertainty) * rnorm(1, 1, model.uncertainty) * rnorm(1, 1, adjust.uncertainty)

We generated three normally distributed random numbers 100,000 times and obtained the confidence intervals from the 100,000 values.

However, the movement of the animal population would cause errors in abundance estimation [41,42]. Therefore, we compared the Tibetan wild ass distribution in summer over different years and checked the stability of its spatial distribution.

## 3. Results

We recorded 159 occurrences of the Tibetan wild ass surveyed in 2017 as example data. The mean group size was 6.5 (the minimum value is 1, and the maximum value is 197), and its standard deviation was 17.8. The total number of observed individuals was 1039, including 103 observations of 449 individuals within the 500 m range (Figure 1). The average detection rate of the distance sampling within 500 m was 64.7% (Figure 2). The detection function can be best-fitted using a uniform function with herm adjustment (a Hermite polynomial term) (Table 1). The actual number of individuals along the survey route was 449/64.7% = 693.7.

Based on the 159 occurrences and 27 environmental variables, the random forest model predicted that Tibetan wild ass abundance was 499,705 individuals (Figure 4). Along the survey route, the predicted abundance was 4585, which was larger by 4585/693.7 = 6.61 times than the number of observed individuals. Therefore, after adjustment, we estimated Tibetan wild ass abundance in the study area to be 75,598. We used the observed animal abundance on the survey route to predict animal abundance in the entire study area. However, most animals were not around the survey route (Figure 6).

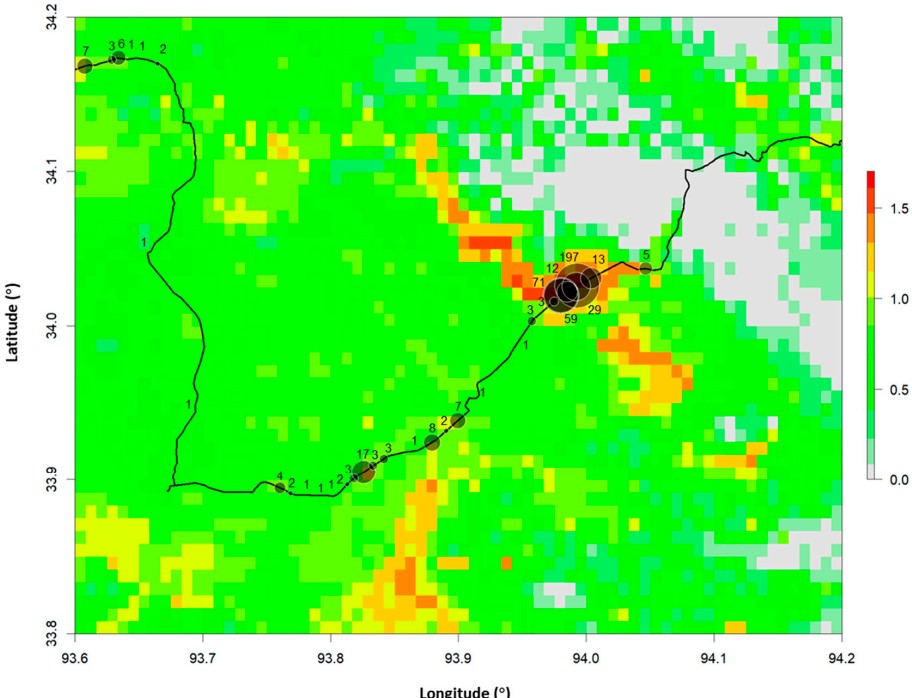

**Figure 6.** The predicted abundance (log-log transformed) of the Tibetan wild ass at Suojia Township, Zhiduo County in the Three-River-Source National Park. The resolution of the abundance prediction map is 1 km². The black circles represent Tibetan wild ass occurrence, and circle size indicates species group size ranging from 1 to 197. The black line is the survey route.

The average detection rate of distance sampling was 64.7%. The $R^2$ of the species distribution model (random forest) was 0.746. The standard deviation of the observation-prediction ratio among the different regions was 0.647. When the survey, model, and adjustment uncertainties were considered, the 95% confidence interval was 13,339–178,215, which is a very wide range. However, when we only consider the uncertainty of the field survey, the species distribution model, or the adjustment, the ranges of the confidence intervals were narrower (Table 2).

**Table 2.** The 95% confidence intervals (CIs) of the estimated animal abundance considering different error sources (field survey, species distribution model, and adjustment).

| Source of Errors | Lower 95% CI | Upper 95% CI | R Code |
|---|---|---|---|
| survey, model, adjustment | 13,339 | 178,215 | CI(EST[4], survey.uncertainty, model.uncertainty, error.adjust) |
| model, adjustment | 22,456 | 144,376 | CI(EST[4], 0.001, model.uncertainty, adjust.uncertainty) |
| survey, adjustment | 15,452 | 158,504 | CI(EST[4], survey.uncertainty, 0.001, adjust.uncertainty) |
| survey, model | 18,283 | 149,844 | CI(EST[4], survey.uncertainty, model.uncertainty, 0.001) |
| adjustment | 28,369 | 123,030 | CI(EST[4], 0.001, 0.001, adjust.uncertainty) |
| model | 37,986 | 113,024 | CI(EST[4], 0.001, model.uncertainty, 0.001) |
| survey | 23,622 | 127,671 | CI(EST[4], survey.uncertainty, 0.001, 0.001) |

We compared the distribution of the Tibetan wild ass in summers over different years based on our surveys, and found its distribution was very stable (Figure S2). Tibetan wild ass populations stay in their territory throughout the year, as they never migrate [43]. In contrast, the Tibetan gazelle (*Procapra picticaudata*) is more mobile than the Tibetan wild ass (Figure S2). The occurrences of the Tibetan fox significantly differed over the four days (Figure S2), and the reason is that it is difficult to detect the fox during distance sampling.

## 4. Discussion

The main features of our method are: (1). using distance sampling to quantify survey uncertainty; (2). applying a SDM (using the random forest algorithm) to predict the animal abundance in the whole study area including unsurveyed regions; (3). adjusting the predicted abundance based on survey results. As such, we can provide a better estimation than traditional methods. Although the N-mixture model can quantify survey uncertainty and the contribution of environmental variables, it is not designed for handling numerous variables to provide a reliable prediction map. Random forest can handle tens of variables (theoretically thousands of variable [38]) and it is convenient for spatial prediction using raster layers. To help researchers to use our method, we produced the abundanceR package to complete the estimation.

We also provided the data of 29 environmental variables covering global terrestrial areas to facilitate the analysis. The data are all publicly accessible, and we standardized their resolution and extension and created a stack file with 29 layers in R. We provided a function to crop the data based on the occurrence of the target animal, and we will upgrade the package to make the analysis more flexible. There are a number of methods using occurrence data to estimate animal abundance. If the home-range and mean group sizes are well established, animal occupancy can be directly scaled up to abundance [25], or a negative binomial distribution method can be used to obtain the global abundance of the target animal [44]. If the contribution of environmental variables is simple, with only linear, quadratic, and interaction effects, animal abundance can be estimated using linear models such as a binomial mixture model [45]. When the animal-environment association is complex with nonlinear, high-order, and interaction effects in a high-dimensional dataset, powerful machine learning algorithms such as random forest are preferable.

To estimate the confidence intervals of the estimated animal abundance, we multiplied the uncertainty of the field survey, the species distribution model, and the adjustment based on the observation-prediction ratio. We considered the uncertainty of field survey because the survey results from direct observation are usually biased [42]. We took into account the uncertainty of adjustment because the variance of detection among sites is often large [46], and the observation-prediction ratio varies in different regions. In the case of the Tibetan wild ass in the Three-River-Source National Park, the range of the confidence intervals is large because the species distribution model only explained half of the variance in animal count at the occurrences [43], and the observation-prediction ratio was different in different regions. If we have the distribution data of wolves and livestock, which strongly influence Tibetan wild ass distribution, the performance of the species distribution model would be substantially improved, and the observation-prediction ratio would be close to one across different regions.

## 5. Recommendations

Our model requires the target species to meet three conditions: (1) the animal lives in an open environment (e.g., grassland, wetland, etc.) where distance sampling is applicable so that the detection rate can be quantified using a detection function, and the uncertainty of the survey can be measured; (2) the species is large and visible (e.g., ungulates) and not elusive (e.g., carnivores); and (3) the target animal is a habitat specialist, so that its distribution is constrained by environmental variables and a properly fitted species distribution model can accurately predict animal abundance in unsurveyed areas. If users can adapt distance sampling for mountain areas [47], such as adjusting study areas from real mountain surface areas to vertical projected areas, they can use our model for mountain species such as the blue sheep (*Pseudois nayaur*) and argali (*Ovis ammon*).

To further minimize errors in our method for wildlife abundance estimation, users can: 1 carry out systematic field surveys covering a good gradient of the animal's habitat; 2 obtain key environmental variables, such as the distribution of predators, preys, and competing species, and various disturbing factors; 3 select animals with stable spatial distributions (avoiding the migration stage) so that survey data from different periods can be combined.

**Supplementary Materials:** The following supporting information can be downloaded at: https://www.mdpi.com/article/10.3390/land11050660/s1. Figure S1. The 29 environmental variables for species distribution modelling. The variables include the 19 climate variables [32], elevation [33], human footprint index [34], and solar radiation, wind speed, and water vapor pressure for January and July, respectively, as well as land cover [35] and wetland [36]. All the 29 layers have the spatial resolution of 1 km². Figure S2. The occurrences and group sizes of the Tibetan wild ass, Tibetan gazelle, and Tibetan fox surveyed on four different days on a 250-km road in the Three-River Source National Park. Table S1. The detection functions, adjustments and values of Akaike information criterion (AIC) in the distance sampling for the Tibetan wild ass at the Three-River Source National Park. Table S2. Variables used in the species distribution model for the Tibetan wild ass in the Three-River Source National Park. The variable names used in the models are in parentheses. The files (BioClim.grd and BioClim.gri) are the dataset of the 29 environmental variables covering the Three-River Source National Park.

**Author Contributions:** Conceptualization, X.L. and E.G.; methodology, X.L. and B.L.; validation, N.L.; formal analysis, X.L. and N.L.; investigation, X.L., B.L. and E.G.; data, X.L. and E.G.; writing—original draft preparation, X.L.; writing—review and editing, X.L., N.L., Y.S. and E.G.; visualization, X.L.; supervision, Y.S.; project administration, X.L.; funding acquisition, X.L. All authors have read and agreed to the published version of the manuscript.

**Funding:** This work was supported by the National Natural Science Foundation of China (No. 31970432 and 32171528), the Second Tibetan Plateau Scientific Expedition and Research Program (STEP, Grant No. 2019QZKK0501), the Third Xinjiang Scientific Expedition Project (Grant No. 2021XJKK1302), the Second National Wildlife Survey Project for Terrestrial Animals and the Key Subject of Ecology of Jiangsu Province (SUJIAOYANHAN(2022) No.2). The study is also supported by Alliance of

International Science Organization (ANSO) (Project ID: ANSO-CR- KP-2020-08) for ecosystem safety assessment and policy making at the China-Mongolia-Russia corridor at the one belt one road zone.

**Institutional Review Board Statement:** Not applicable.

**Informed Consent Statement:** Not applicable.

**Data Availability Statement:** All data used in this study are freely available to anyone. The example data and R code can be accessed at https://github.com/Xinhai-Li/abundanceR. The global environmental variables at the terrestrial areas (a 8 GB compressed file, var29.zip) can be downloaded at https://pan.baidu.com/s/1noU8A7WcsuYx0MSiQq6CeQ (access code is: 1234) for users in Mainland China; and can also be downloaded from the Google Drive at (https://drive.google.com/drive/folders/1bNh4SdikmjrOkgqE5VOVo86SozD2hvmT?usp=sharing).

**Acknowledgments:** We thank staff of Qinghai Forestry Department and the Three-River-Source National Park for their help during the field surveys. We are grateful to Yushan Wang, Ba Zhou, Qianqian Luo, Xiaojia Zhu, Boyi Wang for carrying out the field surveys.

**Conflicts of Interest:** The authors declare that they have no conflict of interest.

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
