# Peer review of "AbundanceR: A Novel Method for Estimating Wildlife Abundance Based on Distance Sampling and Species Distribution Models"

_land, doi:10.3390/land11050660_

Round 1
Reviewer 1 Report
Dear Authors,
I really appreciated your efforts in designing a relatively easy-to-use package to integrate Species Distribution Models and distance sampling in animal abundance estimations. Nevertheless, I found some issues in the way some of the package functions were designed, and in the way you provided the example data (particularly the covariate layers) to replicate the analyses.
Please find below my comments about these 'major' issues, as well as about the minor text-related comments:
Major comments
1) The link provided in Line 151 (https://pan.baidu.com/s/1noU8A7WcsuYx0MSiQq6CeQ) leads to the Chinese version of the website, so it is not easy for a user not being able to read Chinese characters to understand how to download the .zip file. Anyway, I found the place to click on in order to start the download, but then it led me to a .exe file named "BaiduNetDisk_7.12.1.1"; as I can't understand Chinese characters, and I could not find any cue in your manuscript about what this application served for, I did not proceed to its download. Thus, I downloaded the BioClim.rda file from your GitHub page (https://github.com/Xinhai-Li/abundanceR/tree/main/data) and I loaded it in RStudio, but I could not use the resulting RasterBrick object due to denying of permission to access the "D:\GIS\CLIMATE\SJY29.gri" path. Please note that I could instead correctly load and use the kiang.rda and shape.rda files which I downloaded from your GitHub page.
All these words to just tell you that I couldn't use the var29.grd file to test your script. Anyway, I replicated your analyses by cropping the layers of 19 bioclimatic variables downloaded from Worldclim 2.1 repository (at 1-km2 resolution) to the same extent you used in the manuscript (i.e. by using cropLayers(kiang, buffer=0.2, Envlayers=BioClim).
Nonetheless, I suggest you to either replace in the text the https://pan.baidu.com/s/1noU8A7WcsuYx0MSiQq6CeQ link with the one leading to the English version of the web page (if this exists), or to replace it with the link to your GitHub page, provided that you store there a .rda file with the 29 covariate layers accessible to users.
2) There are two major issues in the getEnvData function: (i) as I used only the bioclim layers from Worldclim, I had not the 'elev' layer which is included in the 29 variables you provided in the var.grd file; since the 'elev' layer is explicitly called within the function to remove localities with NAs in covariates (in the command line 'bak = bak[!is.na(bak$elev), ]'), the result of the getEnvData function for me was just a data frame with values for the Bioclim layers at kiang observations. This would happen to anyone who has not a DEM layer named 'elev' in the RasterBrick of covariates. You should either specify this precondition of the getEnvData function, or correct the function itself (e.g. by replacing the above-mentioned command line with something like 'bak = bak[complete.cases(bak),]');
(ii) in case the 'tidyverse' package is loaded, the function will not properly work because the 'extract' function from the 'tidyr' package will supersede that from the 'raster' package.
3) There are other functions in your package which are specifically coded to work with the parameters of your case study, but this compromises a broader use of them: (i) the 'predict' step of the popSize function will smoothly work with model types for which the 'type = response' argument is accepted (RF, but also GLM or GAM) but it would not with Maxent-based SDMs for instance; (ii) you may modify the trackPoints function to let the user possibly set different inter-points intervals (e.g. 100 metres instead of 1 km) along the survey route; (iii) you may modify the estPopSize function to let the user possibly set thresholds for the observer-species distance different than the 500 metres one (which is specifically indicated within the function in the command line 'species = species[species$distance <= 500, ]').
Minor comments
L. 107: Change 'quantify' to 'quantifies' as the subject (the detection function) is singular.
L. 126: Please explain why you used 0.001 as value for the 'convert.units' argument of the ds function.
L. 165-166: The meaning of this sentence is not very clear; to the best of my knowledge, a GLM or a GAM model, with appropriate link function, could predict the number of individuals in every 1-km2 pixel as well, so RF was not the only usable algorithm in this case. Please expand a bit on the reasons of this choice.
L. 188: Here, to make the code smoother, I suggest you to replace 'Dat.fill[, 2:(no.col-5)]' with 'Dat.fill[, 2:ncol(Dat.fill)]'
L. 204: You may use a different color-shape combination for kiang occurrence points (e.g. void blue triangles instead of filled red circles), to let the reader better discriminate between the point and the predicted abundance value at its coordinates in case this latter has a high value (and thus is coloured red based on your chosen scale).
L. 217-218: You may replace "track points" with "trackPoints function".
L. 224-230: this part may be better placed in the Results section; here you may describe with just a couple of sentences how the predicted abundance adjustment process works.
L. 245: I suppose the value 1512 reported here represents a typo error, as above you stated that the adjusted observed individuals should be 449/0.647 = 694.
L. 281: For more clarity about the nature of the "survey_uncertainty" and "model_uncertainty" terms you may add here the R lines generating the corresponding objects, as you did for the error.adjust term.
I guess the command line for survey uncertainty is error.survey = 1 - Average.p (that you show in L. 130), while you only described the model uncertainty term through the wording in L. 270-272.
L. 292: Wasn't the average detection rate 0.647 (L. 129)? The value reported here should be 64.7% in this case.
L. 295-299: Please check the values reported here for total predicted abundance, predicted abundance along the survey route, and adjusted predicted abundance along the route, as they are different from those reported in L. 239-246.
L. 318-321 You should introduce at the end of the Methods the fact that the analyses you made for Tibetan wild ass were made also for Tibetan gazelle and Tibetan fox (please add also the scientific name of this latter) in order to compare the results among these species.
L. 339-340: SDMs may fail in correctly estimating the species-environment relationships also for specialist species if they are not properly tuned, so I will change this sentence with something like '2. the target animal is a habitat specialist, so that its distribution is constrained by environmental variables and a properly fitted species distribution model can accurately predict animal abundance in unsurveyed areas'.
L. 356-357 the fact that your Random Forest model explained only half of the variance, which in turn produced highly variable CI of estimated abundance, may also derive from the parameters you selected (number of trees, size of terminal nodes, number of boostrapped observations used to fit the single trees) not being optimal for your dataset (see for instance Valavi, R., Elith, J., Lahoz‐Monfort, J. J., & Guillera‐Arroita, G. (2021). Modelling species presence‐only data with random forests. Ecography, 44(12), 1731-1742). You should cite this possibility, along with the lack of some possibly relevant predictors that you already cite.
Author Response
Dear Reviewer:
Thank you so much for reviewing our manuscript and provided very pertinent and constructive suggestions. Following your suggestions, we made numerous changes. I listed our responses below. Please see the attached file, which has the same contents yet more clear.
Xinhai Li
Dear Authors,
I really appreciated your efforts in designing a relatively easy-to-use package to integrate Species Distribution Models and distance sampling in animal abundance estimations. Nevertheless, I found some issues in the way some of the package functions were designed, and in the way you provided the example data (particularly the covariate layers) to replicate the analyses.
Please find below my comments about these 'major' issues, as well as about the minor text-related comments:
Major comments
1) The link provided in Line 151 (https://pan.baidu.com/s/1noU8A7WcsuYx0MSiQq6CeQ) leads to the Chinese version of the website, so it is not easy for a user not being able to read Chinese characters to understand how to download the .zip file. Anyway, I found the place to click on in order to start the download, but then it led me to a .exe file named "BaiduNetDisk_7.12.1.1"; as I can't understand Chinese characters, and I could not find any cue in your manuscript about what this application served for, I did not proceed to its download. Thus, I downloaded the BioClim.rda file from your GitHub page (https://github.com/Xinhai-Li/abundanceR/tree/main/data) and I loaded it in RStudio, but I could not use the resulting RasterBrick object due to denying of permission to access the "D:\GIS\CLIMATE\SJY29.gri" path. Please note that I could instead correctly load and use the kiang.rda and shape.rda files which I downloaded from your GitHub page.
All these words to just tell you that I couldn't use the var29.grd file to test your script. Anyway, I replicated your analyses by cropping the layers of 19 bioclimatic variables downloaded from Worldclim 2.1 repository (at 1-km2 resolution) to the same extent you used in the manuscript (i.e. by using cropLayers(kiang, buffer=0.2, Envlayers=BioClim).
Nonetheless, I suggest you to either replace in the text the https://pan.baidu.com/s/1noU8A7WcsuYx0MSiQq6CeQ link with the one leading to the English version of the web page (if this exists), or to replace it with the link to your GitHub page, provided that you store there a .rda file with the 29 covariate layers accessible to users.
Response: Thank you very much for checking the functions and data, and providing suggestions. This is what we really expected and highly appreciated. Now We realized that the Baidu Cloud Disk is only accessible for users in mainland China. As such, we uploaded the file var29.zip to the Google Drive (https://drive.google.com/drive/folders/1bNh4SdikmjrOkgqE5VOVo86SozD2hvmT?usp=sharing), so that it is accessible to most users except those in mainland China. The file var29.zip is a 8 GB file. To make the downloading process easier, I split the file into 12 files. After all the files are downloaded, they can be decompressed to a grd file and a gri file, the total size of the data files is 105 GB. I believe this dataset would be very useful for researchers who run species distribution models. For demonstrating our package, we do not need the data covering the whole planet. We provided another dataset BioClim.grd as a supplementary file, covering the area of the Sanjiangyuan National Park. The file is a 44 MB file. I tried to include it in the package abundanceR, yet when I building the package the source data were not transformed to BioClim.rda, because it is a large file and R only provides the location of the file.
2) There are two major issues in the getEnvData function: (i) as I used only the bioclim layers from Worldclim, I had not the 'elev' layer which is included in the 29 variables you provided in the var.grd file; since the 'elev' layer is explicitly called within the function to remove localities with NAs in covariates (in the command line 'bak = bak[!is.na(bak$elev), ]'), the result of the getEnvData function for me was just a data frame with values for the Bioclim layers at kiang observations. This would happen to anyone who has not a DEM layer named 'elev' in the RasterBrick of covariates. You should either specify this precondition of the getEnvData function, or correct the function itself (e.g. by replacing the above-mentioned command line with something like 'bak = bak[complete.cases(bak),]');
Response: We are sorry that we didn’t appropriately provide the data (either the global data var29.grd or the local data BioClim.grd) in the first version. Now both datasets are available. We suggest users to use the datasets we provided, which including 29 layers of environmental variables. The global dataset covers the whole terrestrial areas on the planet at the resolution of one square km, so that it is suitable for most cases. Currently the functions of abundanceR were designed to use the specific datasets. In the next step, we will make it suitable to other data types.
(ii) in case the 'tidyverse' package is loaded, the function will not properly work because the 'extract' function from the 'tidyr' package will supersede that from the 'raster' package.
Response: Thank you so much for reminding us of this issue. We used raster:: extract() instead in the revised code.
3) There are other functions in your package which are specifically coded to work with the parameters of your case study, but this compromises a broader use of them: (i) the 'predict' step of the popSize function will smoothly work with model types for which the 'type = response' argument is accepted (RF, but also GLM or GAM) but it would not with Maxent-based SDMs for instance; Yes, the functions are specifically coded to work with the parameters of our case study. We will gradually upgrade the package to make it more general. Maxent is a very popular algorithm for species distribution modelling, and we will include it at the next step. (ii) you may modify the trackPoints function to let the user possibly set different inter-points intervals (e.g. 100 metres instead of 1 km) along the survey route; The 1 km interval is to match the environmental variables at 1 square km resolution. In the next step, we will open to use other type of environmental variables, and the inter-points intervals will be adjusted accordingly. (iii) you may modify the estPopSize function to let the user possibly set thresholds for the observer-species distance different than the 500 metres one (which is specifically indicated within the function in the command line 'species = species[species$distance <= 500, ]'). The 500 meters threshold was also set to match the 1 square km resolution, as the total distance on the two sides of the road add up to 1000 m. Whenever the resolution of the environmental variables changes, the threshold will be changed accordingly.
Minor comments
- 107: Change 'quantify' to 'quantifies' as the subject (the detection function) is singular.
Response: The error was corrected following the suggestion.
- 126: Please explain why you used 0.001 as value for the 'convert.units' argument of the ds function.
Response: In the data table (kiang), the distance is in metres, and the area in the analysis (providing population density per square km) is in square kilometres, so that we used convert.units=0.001.
- 165-166: The meaning of this sentence is not very clear; to the best of my knowledge, a GLM or a GAM model, with appropriate link function, could predict the number of individuals in every 1-km2 pixel as well, so RF was not the only usable algorithm in this case. Please expand a bit on the reasons of this choice.
Response: In our package, we only use random forest to predict the number of individuals in every 1-km2 pixel. Because random forest usually had better performance than GLMs or GAMs. Random forest is a machine learning algorithm that fits well the high order and interaction effects of the explanatory variables, and it is especially suitable for species distribution modeling.
- 188: Here, to make the code smoother, I suggest you to replace 'Dat.fill[, 2:(no.col-5)]' with 'Dat.fill[, 2:ncol(Dat.fill)]'
Response: This is a very good suggestion that it actually pointed out an error. We corrected the error based on the suggestion.
- 204: You may use a different color-shape combination for kiang occurrence points (e.g. void blue triangles instead of filled red circles), to let the reader better discriminate between the point and the predicted abundance value at its coordinates in case this latter has a high value (and thus is coloured red based on your chosen scale).
Response: That is a good suggestion. We changed the color from red to blue.
- 217-218: You may replace "track points" with "trackPoints function".
Response: Thanks. Changed as suggested.
- 224-230: this part may be better placed in the Results section; here you may describe with just a couple of sentences how the predicted abundance adjustment process works.
Response: We moved this paragraph to the result section, and adjusted the sentences to fit the context.
- 245: I suppose the value 1512 reported here represents a typo error, as above you stated that the adjusted observed individuals should be 449/0.647 = 694.
Response: Thank you so much to finding this error. This value was based on our previous analysis, which was out of date..
- 281: For more clarity about the nature of the "survey_uncertainty" and "model_uncertainty" terms you may add here the R lines generating the corresponding objects, as you did for the error.adjust term.
I guess the command line for survey uncertainty is error.survey = 1 - Average.p (that you show in L. 130), while you only described the model uncertainty term through the wording in L. 270-272.
Response: We added two lines in the menuscript, giving the code for calculating the values of survey uncertainty and model uncertainty. We also changed the names of “error.survey” to "survey.uncertainty", changed the names of “error.adjust” to "adjust.uncertainty", making the three uncertainties consistent.
The added code is:
# survey.uncertainty = 1 - Average.p # survey uncertainty
# model.uncertainty = 1-max(RF$rsq) # model uncertainty
- 292: Wasn't the average detection rate 0.647 (L. 129)? The value reported here should be 64.7% in this case.
Response: Yes, it should be 64.7%. We have adjusted the data and model parameters for many times, the value 80.9% was based on another survey route.
- 295-299: Please check the values reported here for total predicted abundance, predicted abundance along the survey route, and adjusted predicted abundance along the route, as they are different from those reported in L. 239-246.
Response: The values are wrong and we correct them. We are sorry for such low-level errors.
- 318-321 You should introduce at the end of the Methods the fact that the analyses you made for Tibetan wild ass were made also for Tibetan gazelle and Tibetan fox (please add also the scientific name of this latter) in order to compare the results among these species.
Response: We did not do the same estimation for the abundance of the Tibetan gazelle and Tibetan fox, because they are more mobile or hard to detect. We compared the distribution patterns of the three animals in order to clarify what is the suitable target animals (such as the Tibetan wild ass) for our package.
- 339-340: SDMs may fail in correctly estimating the species-environment relationships also for specialist species if they are not properly tuned, so I will change this sentence with something like '2. the target animal is a habitat specialist, so that its distribution is constrained by environmental variables and a properly fitted species distribution model can accurately predict animal abundance in unsurveyed areas'.
Response: Thank you so much for the suggestion. A well-fitted model is an important condition, so that we accept this suggestion, and removed the next sentence “As such, the species distribution model can accurately predict animal abundance in unsurveyed areas”.
- 356-357 the fact that your Random Forest model explained only half of the variance, which in turn produced highly variable CI of estimated abundance, may also derive from the parameters you selected (number of trees, size of terminal nodes, number of boostrapped observations used to fit the single trees) not being optimal for your dataset (see for instance Valavi, R., Elith, J., Lahoz‐Monfort, J. J., & Guillera‐Arroita, G. (2021). Modelling species presence‐only data with random forests. Ecography, 44(12), 1731-1742). You should cite this possibility, along with the lack of some possibly relevant predictors that you already cite.
Response: The suggested paper is an excellent paper, and I learned a lot from it. I tried the Equal-sample RF and the Down-sampled RF, but the performance was worse than our model. I realized that our data are count data, not presence data, so that the model is similar to a weighted model, with points with large numbers of individuals being weighted higher. We cited this paper in our manuscript.
Reviewer 2 Report
General comments:
Description of distance sampling and SDMs lack key information. It is very difficult to judge about the validity of the approaches used.
All used R packages must be cited! I stumbled across some missed citations, e.g., “RandomForest”
The methods section gives too many results, which are difficult to interpret at this point; I suggest revising and strictly dividing between describing the method and already showing actual data
The discussion of virtually absent, only a very short paragraph. This must be improved.
Sections (Intro, methods, results) are mixed all over and must given where they belong
Very interesting idea, but as a methodological description, I recommend using a better dataset or simulated data. Also, the documentation of data and all steps must be more carefully reported. Please see comments below.
Specific comments:
Abstract
L19 the new sentence stands a bit alone here. Can you better connect to the previous? Maybe like “in contrast to that…”
The manuscript cannot hold what the abstract is promising. e.g., L32f how accurate are the values?
Introduction
The first two paragraphs jump a bit back and forth. You start with abundance estimation and the introduce SDMs, then jump from small to large scale and back. Maybe start with SDMs working on larger scales with a few examples and then zoom in to opportunities from small-scale studies: the abundance estimation
Methods
The introductory paragraph is not needed. It repeats information given elsewhere in the text or things that could be more easily included further downstream
Paragraph “Date requirements”: I would rather call it dataset or exemplary data. Please indicate the sample size (how many sighting locations or animals). Or maybe split into study region plus environmental data and on section about the survey protocol etc. How often did you survey the region? once? Several times? Must be stated.
L123 why was this truncation range chosen? This must be explained here already
Figure 2: What was the number of sightings used? Approx. 23? How did you treat clusters/groups of animals? This must be explained somewhere. Buckland et al. (2001) say that n should be 60-80 observation for reliable estimates (page 240 in my edition of the book). Around 40 may be the lowest amount of observations, but less is not reliable. You must be more specific here! The detection function is not really a good fit to the data…
Section 2.3: I still think this can be combined with the study region information. Did you check for multicollinearity among the 29 variables? If yes, this must be written out, if not, this must be conducted. Another option would be to transform the 29 explanatory variables to principal components before the SDMs. But then no temporal extrapolation is possible.
L164: if “MARS” does not appear again in the manuscript, there is no need to provide the abbreviation; may distract readers
The argument for selecting random forests and not testing the others is too weak here. If it is mainly because of the use of abundance as model output, this must be highlighted more, and stated that other model don’t provide this opportunity. Combined with the following sentence, I am a little confused. What is the configuration of the response variable? Presence-absence? Or Absence versus several abundance categories. Must be clarified! Also, was abundance data standardized to 1km² grid cells? If so, this must be stated much more prominent!
L180: it must be further explained why you used na.roughfix. Not clear to me.
L194: how as landcover and wetland configurated? Landcover in categories and wetland as binary? This must be stated; or maybe as you actually dropped these variables, you may not include them at all in the modeling
Figure 4: here it becomes apparent that not knowing about the sample size of sightings makes it impossible to judge about the validity.
L223ff: You are presenting numbers here of abundances and detection rates. I think these are means of estimates. In all Distance sampling literature, Capture-capture, SECR etc. You always report confidence intervals, sometimes also coefficients of variation standard errors.. So you should also make your method scientifically accurate! You are multiple figures by others without telling about the accuracy of the estimates. If confidence intervals were already high in the first calculation, you sum up uncertainty. Like this you don’t even report the amount of uncertainty. This must be changed! To make it more clear, your calculation looks like done with raw model outputs in a quick-and-dirty way. This must be scientifically robust.
L263ff: are there any references that also use 1-detection rate as a measure of uncertainty? I would rather look for that in the width of confidence intervals.
L278: What is “a certain amount of standard deviation”? SD is a measure of variance around the mean. This must be re-written and clarified
L285: With what kind of data? Did you repeat the survey over several years? Did you use other data? Can you rule out other effects that could influence the population size? Migration, mortality, weather effects on sampling days?! This is completely unclear to me!
Results
L289: sample size must go to methods section
L290: “The mean group size was 6.5±17.8 SD” is a commonly used way of descriptive statistics. In cases where the SD is larger than the estimate for the mean, you may also report min/max values and the median, as the distribution seems to be highly skewed.
L291ff: You give data here, but the figures 1 and 2 don’t actually show this exact data, they somehow show something related. Links between text and figures must be improved!
L296: this remarkable mismatch can be almost entirely due to survey effort. Did you drive the route once? Then I would attribute the mismatch to that. Or did you repeat surveys? All of this must be specified in the methods! It is impossible to judge about the robustness of your estimates!
L310: I don’t get why you use the combined uncertainty measure when it goes form negative six to almost 300. If this formula is valid, the uncertainty can be virtually everything. Here you already start interpreting the results. This must go to the discussion!
L316: what kind of data is that now? From literature? Own data? It must be properly described, cited etc. This paragraph may further belong to the discussion.
Discussion
L329: it actually can, if you have comparable habitat characteristics and a defined survey area.
L333: This information should be places in the introduction, as it opens up why this study is actually needed. If you applied the idea from a different approach, this must be clearly stated beforehand, and not in the discussion for the first time.
Until L342: this is introduction material; I suggest moving it there. You could start with a short introductory paragraph and then discuss your method and results.
L343-351: still introduction
L352: now the discussion starts.
L363ff: if all these factors could still influence your method, how can you judge about its validity and describe it as something worth using in other studies?
L370ff: this is repeating the methods section here again. Where is the actual discussion of your method? Despite saying that it cannot be judged about it because of weaknesses in the dataset, you don’t give any further viewpoints on it. You end with stating this method could be useful for other researchers. In the current form, I don’t think so. The key idea is likely worth following, but you should rather start with a good dataset or simulated data to prove that your method is working. Then you could apply it to real world data.
Table S1: please also provide delta AIC values; that makes it easier for the reader to see the differences in model performance
Figure S2: this is also your own data? Is it worth reporting illustrating the use of your new method? This is furthermore the first time I read about the survey effort… Must be given in methods!
Author Response
Dear Reviewer:
Thank you so much for taking time to review our manuscript, and provided comments and suggestions. Following your suggestions, we made numerous changes. I listed our responses in red for clarity.
Xinhai Li
General comments:
Description of distance sampling and SDMs lack key information. It is very difficult to judge about the validity of the approaches used.
Response: To make the roles of distance sampling and SDMs clear, we added “The advantage of distance sampling is that it estimates the detection function, quantifying the relationship between the probability of detection and the animal-to-observer distance, so that it provides a measure of survey uncertainty.”, and for SDMs “…, as they can quantify the association of species occurrences and environmental variables” in the introduction section. Please see lines 52-57.
We did not specifically give the definitions of distance sampling and SDMs in the manuscript. The manuscript proposes a new R package for estimating animal abundance, and the potential readers are ecologists, who are aware of distance sampling and SDMs, and they will use this package for their own studies. People without the background of distance sampling and SDMs have no interests in applying our package, so that we did not focus on these readers. Anyway, biologists can understand the rationale of our package from the context.
All used R packages must be cited! I stumbled across some missed citations, e.g., “RandomForest”
Response: Thanks for the suggestion. We added the citation for “randomForest”. Other packages we used are Distance, devtools, maptools, raster, rgdal, and sp. We already provided the citation for Distance. The next five packages were not mentioned in the context, so that we did not give their citations.
The methods section gives too many results, which are difficult to interpret at this point; I suggest revising and strictly dividing between describing the method and already showing actual data
Response: We deleted the paragraph “As an example, …” in the section 2.5 Adjusting model prediction, as it is the results of the study. The relevant contents were combined into the result section.
The discussion of virtually absent, only a very short paragraph. This must be improved.
Response: I think there is some misunderstanding here. We have six paragraphs in the discussion section, including the comparison with other methods for abundance estimation, the key features of our package, the weakness of our method, and some useful hints for R users. We also listed five points that can minimize estimation errors.
Sections (Intro, methods, results) are mixed all over and must given where they belong
Response: We agree some contents were mixed, and we made adjustments in this revision.
Very interesting idea, but as a methodological description, I recommend using a better dataset or simulated data. Also, the documentation of data and all steps must be more carefully reported. Please see comments below.
Response: The dataset of the Tibetan wild ass is based on my own surveys. The surveys for the ass triggered me to develop this method. The dataset is not perfect, yet it is real and reflects the reality of the uncertain nature. It is a good dataset for testing our method.
Specific comments:
Abstract
L19 the new sentence stands a bit alone here. Can you better connect to the previous? Maybe like “in contrast to that…”
Response: Species distribution models are parts of previous mentioned methods, so that “in contrast to that…” cannot be used here. An ecologist would feel the logic smooth because species distribution models are deeply involved in studying the distribution and abundance of animals. When we mentioned distribution or abundance estimation, species distribution models would be naturally thought.
The manuscript cannot hold what the abstract is promising. e.g., L32f how accurate are the values?
Response: We have very high confidence that we provided much accurate estimation of animal abundance for certain species. We filled the research gap that species distribution models can hardly be applied to estimate animal abundance, and we clarified the constrains of our method (only good for open area animals which are restricted by environmental variables).
Introduction
The first two paragraphs jump a bit back and forth. You start with abundance estimation and the introduce SDMs, then jump from small to large scale and back. Maybe start with SDMs working on larger scales with a few examples and then zoom in to opportunities from small-scale studies: the abundance estimation
Response: In the first paragraph, we have the top sentence: “Species abundance is fundamental information for ecological research, and various methods are used for its estimation [1-3].”. We listed the methods following the top sentence. The logic is very simple.
To connected with the second paragraph, we mentioned that the methods, mark-recapture and camera traps, are appropriate for small regions. As such, the next paragraph was naturally followed for large regions.
Our method is for a large-scale estimation of the animal abundance. We cannot narrow down our method to small-scale studies.
Methods
The introductory paragraph is not needed. It repeats information given elsewhere in the text or things that could be more easily included further downstream.
Response: This paragraph is a key part introducing the whole picture of the method section. Otherwise, the readers would start from the details and lose the whole picture.
Paragraph “Date requirements”: I would rather call it dataset or exemplary data. Please indicate the sample size (how many sighting locations or animals). Or maybe split into study region plus environmental data and on section about the survey protocol etc. How often did you survey the region? once? Several times? Must be stated.
Response: Thanks for the suggestion. We replaced “Date requirements” by “Exemplary data”. No specific sample size is required, and no repetition of survey is needed. In short, a few occurrences from one survey can be used by our package for abundance estimation, but it would not be very meaningful because the confidence intervals would be very large.
L123 why was this truncation range chosen? This must be explained here already
Response: This point had been clarified in the paragraph below Fig. 2. We moved the paragraph in front of the R code. The paragraph is: “The maximum distance that the Tibetan wild ass can be detected is 1,500 m, yet we set the truncation distance to 500 m, i.e., the animals outside the range were ignored. As such, the detection range on each side was 500 m, and the width of the line transect was 1 km; thus, the field observations were comparable with the model prediction at the same spatial scale, i.e., 1-km2 quadrats.”.
Figure 2: What was the number of sightings used? Approx. 23? How did you treat clusters/groups of animals? This must be explained somewhere. Buckland et al. (2001) say that n should be 60-80 observation for reliable estimates (page 240 in my edition of the book). Around 40 may be the lowest amount of observations, but less is not reliable. You must be more specific here! The detection function is not really a good fit to the data…
Response: There are 103 groups/occurrences of the wild ass being used in this figure. We added the number in the result section. The animals were not evenly distributed, and we did not transform the data. Although the number of observation is larger than 60-80, we agree the detection function is not a good fit to the data.
Section 2.3: I still think this can be combined with the study region information. Did you check for multicollinearity among the 29 variables? If yes, this must be written out, if not, this must be conducted. Another option would be to transform the 29 explanatory variables to principal components before the SDMs. But then no temporal extrapolation is possible.
Response: We designed a package which is suitable for any terrestrial region on the planet, so that we did not combine the environmental variables to any region. Some of the variables are highly correlated, yet the multicollinearity is never a problem for random forest. Principal component analysis is an option for linear models, yet random forest way overfit linear models, so that we did not use linear models. In this study, we did not carry out temporal extrapolation, but spatial extrapolation.
L164: if “MARS” does not appear again in the manuscript, there is no need to provide the abbreviation; may distract readers
Response: Agree. We removed the abbreviation.
The argument for selecting random forests and not testing the others is too weak here. If it is mainly because of the use of abundance as model output, this must be highlighted more, and stated that other model don’t provide this opportunity. Response: The good performance of random forest has been proved by numerous studies. We cited four reprehensive papers for support (line 178). Well, predicting animal abundance can be achieved by many models, so that I removed the sentence from here. Combined with the following sentence, I am a little confused. What is the configuration of the response variable? Presence-absence? Or Absence versus several abundance categories. Must be clarified! Also, was abundance data standardized to 1km² grid cells? If so, this must be stated much more prominent!
Response: This is the key question. I am sorry that I did not make it very clear. We added: “The dependence variable is the count of the Tibetan wild ass along the survey routes and evenly distributed pseudo-absence data (count 0) in the study area (Figure 3).”.
L180: it must be further explained why you used na.roughfix. Not clear to me.
Response: We added: “, because in certain places some variables such as human footprint index has null values while other variables have valid values”.
L194: how as landcover and wetland configurated? Landcover in categories and wetland as binary? This must be stated; or maybe as you actually dropped these variables, you may not include them at all in the modeling
Response: The landcover and wetland are all categorical variables. We did not use the two variables in our study. However, the two variables are important for some other species such as water birds. I’d like to provide such data to facilitate our researchers’ work.
Figure 4: here it becomes apparent that not knowing about the sample size of sightings makes it impossible to judge about the validity.
Response: We revised the statement as: “The red circles indicate the 159 Tibetan wild ass occurrences s of 1,039 individuals, …”. I repeated the surveys for four time in four years, and the distribution pattern of the ass remained. I mean the presence and absence of the wild ass along the survey routes is reliable.
L223ff: You are presenting numbers here of abundances and detection rates. I think these are means of estimates. In all Distance sampling literature, Capture-capture, SECR etc. You always report confidence intervals, sometimes also coefficients of variation standard errors.. So you should also make your method scientifically accurate! You are multiple figures by others without telling about the accuracy of the estimates. If confidence intervals were already high in the first calculation, you sum up uncertainty. Like this you don’t even report the amount of uncertainty. This must be changed! To make it more clear, your calculation looks like done with raw model outputs in a quick-and-dirty way. This must be scientifically robust.
Response: The whole paragraph (lines 223-231) was deleted. As to the uncertainty of abundance estimation, we combined the survey uncertainty based on detection rate of distance sampling, the model uncertainty based on the R square of the random forest model, and the adjust uncertainty based on the spatial heterogeneity of model bias. Fully quantifying the estimation uncertainty is the strength of our study.
L263ff: are there any references that also use 1-detection rate as a measure of uncertainty? I would rather look for that in the width of confidence intervals.
Response: I tried to find a reference but failed. We assume the high detection rate means low uncertainty. We actually expect further discussion here. We shared all the data and formula for criticization.
L278: What is “a certain amount of standard deviation”? SD is a measure of variance around the mean. This must be re-written and clarified
Response: This statement did not make sense. We changed to “We assumed that the three indices of uncertainty are independent. Therefore, we multiplied them together as the overall uncertainty:
R code.
L285: With what kind of data? Did you repeat the survey over several years? Did you use other data? Can you rule out other effects that could influence the population size? Migration, mortality, weather effects on sampling days?! This is completely unclear to me!
Response: We used data from distance sampling from 2014 to 2015. The distribution patterns were same, so that the effects of migration, mortality, and weather can be ruled out. Here is the method section, so we did not provide all the information. We showed the results in Figure S2.
Results
L289: sample size must go to methods section
Response: We agree that the sample size should be provided in the method section. We repeated this number here in order to smoothly bring other numbers based on our analysis.
L290: “The mean group size was 6.5±17.8 SD” is a commonly used way of descriptive statistics. In cases where the SD is larger than the estimate for the mean, you may also report min/max values and the median, as the distribution seems to be highly skewed.
Response: Thanks. We added “(the minimum value is 1, and the maximum value is 197)” after “The mean group size was 6.5”. The large variance is because we met a large group with 197 individuals, and another second large group with 97 individuals.
L291ff: You give data here, but the figures 1 and 2 don’t actually show this exact data, they somehow show something related. Links between text and figures must be improved!
Response: Thanks for the suggestion. We added the numbers to the figure captions in Figure 1 and Figure 4.
L296: this remarkable mismatch can be almost entirely due to survey effort. Did you drive the route once? Then I would attribute the mismatch to that. Or did you repeat surveys? All of this must be specified in the methods! It is impossible to judge about the robustness of your estimates!
Response: Such mismatch is very common, so this is why Mark Boyce at al. (2016) claimed that species distribution can not be used for abundance estimation. We made effort to quantify such mismatch and adjusted the abundance accordingly. I personally surveyed the study area for four times (2014-2017), and the distribution pattern of the wild ass remained unchanged, so that the observation-prediction ratio is stable.
Boyce, M.S.; Johnson, C.J.; Merrill, E.H.; Nielsen, S.E.; Solberg, E.J.; van Moorter, B. Can habitat selection predict abundance? J. Anim. Ecol. 2016, 85, 11–20,
L310: I don’t get why you use the combined uncertainty measure when it goes form negative six to almost 300. If this formula is valid, the uncertainty can be virtually everything. Here you already start interpreting the results. This must go to the discussion!
Response: Using the combine uncertainty is the most conservative way (maximize the errors). The statement “which is a very wide range” is a plain description of the result. Following this statement, we provided the confidence intervals based on single-factor uncertainty. We feel the statement can stay in the result section.
L316: what kind of data is that now? From literature? Own data? It must be properly described, cited etc. This paragraph may further belong to the discussion.
Response: Thanks for the questions. We did not make it clear. We changed the sentence to: “We compared the distribution of the Tibetan wild ass in summers over different years based on our surveys, and found its distribution was very stable (Figure S2).”.
Discussion
L329: it actually can, if you have comparable habitat characteristics and a defined survey area.
Response: It can extend to unsurveyed areas only under the assumption that the habitat characteristics are same. However, the habitats are highly spatially heterogeneous. Furthermore, I surveyed the Mongolia grassland, where the landscape looked homogenous in a very large area, yet the animal distribution is highly heterogeneous.
L333: This information should be places in the introduction, as it opens up why this study is actually needed. If you applied the idea from a different approach, this must be clearly stated beforehand, and not in the discussion for the first time.
Response: Our method is needed because it filled the gap: distance sampling cannot extend to unsurveyed area, whereas SDMs can but have bias. Our method can adjust the bias and provide reliable estimation. Here, in the discussion section, we made broader comparisons with other methods. The N-mixture model is a simple model that not designed for spatial prediction (not like SDMs), so that we did not introduce it in the introduction section.
Until L342: this is introduction material; I suggest moving it there. You could start with a short introductory paragraph and then discuss your method and results.
Response: In the introduction section, we already mentioned about the mark-recapture method. Here, I discussed the difference of our method with others. We defined a specific position for our method (conditions for using the method) in this paragraph. The sentence in line 342 followed the previous sentence about large areas.
L343-351: still introduction
Response: This is a further discussion about the position of our method. Without the method section and result section, how can we compare the technical features of our model with other models? We feel it very hard to move it to the introduction section.
L352: now the discussion starts.
Response: Here we talk about the confidence intervals. This is the last part we need to discuss.
L363ff: if all these factors could still influence your method, how can you judge about its validity and describe it as something worth using in other studies?
Response: What the factors influenced is not our method, it is the estimated abundance. Estimating wildlife abundance always involved uncertainty. What we did it to quantify the uncertainty. Such uncertainty maybe large for some surveys, maybe small for others, and it is not our concern. We care about the justice of our method, not specific results and associated uncertainty.
L370ff: this is repeating the methods section here again. Where is the actual discussion of your method? Despite saying that it cannot be judged about it because of weaknesses in the dataset, you don’t give any further viewpoints on it. You end with stating this method could be useful for other researchers. In the current form, I don’t think so. The key idea is likely worth following, but you should rather start with a good dataset or simulated data to prove that your method is working. Then you could apply it to real world data.
Response: In the last paragraph of our manuscript, we repeated that we shared a very good dataset. As a senior modeler, I deeply realized that many researchers struggle to look everywhere for data. I made great effort to compile the global data of 29 variables at fairly fine scale (1 km resolution), and I publicly share the data, which is enough for most species distribution models. I gave a hint to remind readers not to forget this offer. You can say my dataset has weaknesses, but it is real. I don’t think a perfect fake dataset can do anything better when using our method. We developed an R package, and it is ready for all researchers. The example dataset, the code of our model, are all freely accessible. We are open to take criticization.
Table S1: please also provide delta AIC values; that makes it easier for the reader to see the differences in model performance
Response: We added the delta AIC values.
Figure S2: this is also your own data? Is it worth reporting illustrating the use of your new method? This is furthermore the first time I read about the survey effort… Must be given in methods!
Response: This is my own data. We mentioned it in the last paragraph of the method section, only used for proving the spatial distribution of the Tibetan wild ass is stable. It is not about survey effort.
Reviewer 3 Report
I have read with interest this work as this method is in my opinion an important contribution to species distribution modelling. The combination of SDM and abundance is very useful not only for conservation purposes but also for scientific works. Altough I clearly think this work is publishable in Land, I think the introduction should be expanded and improved. The relevance of the method needs a better and more extensive introduction, with a better contextualization. In addition, I must say that the mathematic basis of the algorithms should be revised by an expert (mathematical or statistical). I trust in the suitability of the matemathical method, but I do not think cualified to judge it.
For these reasons I recommend major revisions, and I wish the authors luck
Author Response
Dear Reviewer:
Thank you very much for taking time to review our manuscript and providing your comments. We have revised the manuscript following the comments and suggestions. I inserted the point-to-point responses in red.
Xinhai Li

Reviewer 4 Report
1.The Discussion section needs to be re-written and expanded to include a critic evaluation of their results and interpretation of them in relation to the published literature (incluiding their own, similar research, e.g. https://www.sciencedirect.com/science/article/abs/pii/S1574954122000462?via%3Dihub). The authors are considering that all their algorithms are valid and they do not discuss them, beyond of stablishing some limitations for their use.
2. The authors should consider to run/evaluate the effect on the population estimates of some of the determining parameters. This will improve the ms because it will help to assess the vulnerability of their proposed methodology.
3. One of the requirements of this methodology is homogeneity of the area, however authors do not explain how and why the lack of homogeneity might affect the estimates.
4. Other comments: in relation to what they say in Line 360 :
We produced the abundanceR package to complete the estimation. This method requires two conditions: 1. the animal lives in an open environment where distance sampling is applicable so that the detection rate can be quantified using a detection function, and the uncertainty of the survey can be measured,
the following paper that might be used to correct this deficiency for animals living in mountains.
https://digital.csic.es/handle/10261/22242
Author Response

(The authors gave the same response as above.)
